# Comparative Analysis of Long-Term Care in OECD Countries: Focusing on Long-Term Care Financing Type

**DOI:** 10.3390/healthcare11020206

**Published:** 2023-01-10

**Authors:** Seok-Hwan Lee, Yongho Chon, Yun-Young Kim

**Affiliations:** 1Seoul50plus Foundation, Seoul 04147, Republic of Korea; 2Department of Social Welfare, Incheon National University, Incheon 22012, Republic of Korea; 3Department of Social Welfare, Jeonbuk National University, Jeonju 54896, Republic of Korea

**Keywords:** long-term care, long-term care financing, Gilbert and Terrell’s policy analysis framework

## Abstract

This study aims to examine the characteristics of long-term care (LTC) financing in Organization for Economic Cooperation Development (OECD) countries. To this end, the 26 OECD countries that have introduced LTC systems were classified into three types of models: tax-based, health insurance, and LTC insurance (LTCI) systems. Thereafter, these systems were analyzed using Gilbert and Terrell’s policy analysis framework. The results indicated differences in the characteristics of each type of financing in terms of allocation, benefit provision, service delivery, and finance. It is likely that while the LTC insurance model was mainly based on universalism and showed the highest level of coverage, the tax-based and health insurance models adopted selectivism with lower level of benefits per capita. In terms of service delivery, local authorities tended to have the responsibility to decide LTC service users and provide services in many countries, regardless of the type of model. In terms of finance, LTC insurance-based countries had the highest LTC expenditure as a percentage of GDP, followed by countries with tax-based and health insurance systems.

## 1. Introduction

The socialization of care has become a major policy in most OECD countries due to new social risks associated with low fertility and aging. In particular, the LTC system was introduced to respond to the care crisis of the elderly following the retirement of baby boomers as a representative de-familiarization policy [1]. Based on OECD statistics [2], as of 2018, the average LTC expenditure as a percentage of the GDP of 17 member countries was 1.47%, and the ratio is expected to increase by up to three times by 2050. The increasing demand for LTC for the elderly and accompanying financial pressures have led to institutional reforms in most countries that have introduced LTC systems [3]. The reason this study started with data from the 2010s is that it represents the results after institutional reform in the 1990s.

The general direction of reforms appears to be the de-institutionalization and re-familiarization of care. Traditionally, LTC has been mainly provided by family members, especially women. However, it has been carried out by the state and market through the policy of socialization of LTC in advanced welfare states. In addition, it shifted from institutionalized care to community care to support aging in place for the elderly [4].

The financing system significantly determines the design of an LTC system and the contents of its services, which is based on differences in the path dependency of each country’s social security, health insurance systems, and so on [5]. However, few studies have examined the characteristics of institutional design and development processes based on the financing systems in OECD countries and differences that emerge in terms of services.

The purpose of this study is to determine major characteristics by the finding commonalities and differences in each type of LTC financing system. This study is meaningful since it classifies the financing system of OECD countries into tax-based, health insurance, and LTC insurance (LTCI) and broadens the understanding of the system by analyzing the characteristics of each subtype. The analysis target was 26 countries that belong to the OECD. The countries were classified according to their financing system, such as tax, health insurance, and LTCI. They were then analyzed using the policy analysis framework in terms of allocation, benefit provision, delivery, and finance proposed by Gilbert and Tarrell [6]. Through this, we draw implications from the characteristics according to the type of financing.

## 2. Theoretical Background: Long-Term Care System and Financing System

The financing system adheres to the welfare state regime and historicity. Financing systems for LTC in OECD countries can be classified into tax-based, social insurance-based, and private insurance-based models, and mixed models are common [7]. Social insurance is classified into health insurance-based and LTC insurance-based systems. A majority of countries are operating a tax-based LTC system. For instance, Nordic countries, such as Sweden, Denmark, Norway, and Finland; Eastern European countries, such as Austria, Czech Republic, Poland, Slovakia, and Slovenia; Southern European countries, such as Portugal and Spain; and Western European countries, such as the UK, Ireland, France, Australia, and Canada are operating a tax-based LTC system. Meanwhile, European countries such as Estonia, Hungary, Belgium, Switzerland, and Poland are operating a health insurance system. Only a few countries, such as Germany, the Netherlands, Luxembourg, Japan, and Republic of Korea have an LTC insurance system. Meanwhile, most countries with social insurance as their main source of finance, such as health insurance and LTC insurance, supplement their finances through taxation. Private insurance has grown remarkably in countries such as France and Germany, but the scale is small and not universalized [8].

The LTC system is connected to the existing social security system in each country to form a financial system. From a publicity perspective, several OECD countries perceive long-term care services as universal services that are paid by taxation [2].

However, countries with social insurance systems, like Germany, Republic of Korea, Japan, Luxembourg, and the Netherlands, also offer long-term care insurance. It is a system in which people pay premium regularly like a pension. These countries pursued the continuity of the welfare state while reducing financial burden due to taxation, expanding the recipients of long-term care services, expanding the diversity of benefits, and expanding supply in the private sector. In contrast, the United States, Belgium, and Hungary are operating long-term care systems within health insurance, and they are also promoting the role of the private sector [8].

Financial resources have been consistently addressed in existing LTC system researches. In particular, since the introduction of the LTCI system, an interest in the sustainability of the LTC system has increased as the financial burden caused by low birth rates and aging has worsened. Accordingly, the majority of studies have explored effective countermeasures [7,9,10,11,12]. However, only a few studies have analyzed the differences between countries according to their LTC financing system. Previous studies have examined one country with one financing type [13,14] or have conducted comparative studies between countries [15], regardless of the system, and have focused only on the financing method [16,17].

## 3. Research Method

This study used the policy analysis framework proposed by Gilbert and Terrell [6] to compare the characteristics of the financing systems of 26 OECD countries that have introduced LTC systems. Gilbert and Terrell’s framework [6] has been commonly used by scholars to compare the systems and outcomes of the policies of different countries because it suggested a useful framework to classify and analyze essential elements of policy design into four areas, including allocation, benefits, delivery system, and finance. These are defined given the aim of the study as follows (Table 1). Allocation means to whom LTC service will be provided. Selectivism is based on the eligibility criteria of means (or/with income) and the degree of need of care (such as the Activities of Daily Living). It tends to provide public-based, long-term services, mainly to the poor. However, universalism is only based on the degree of the need of care, regardless of the financial situation of the person.

Benefit refers to the method and degree of service, and it was examined in terms of the ratio of home and institution benefits and the amount of per capita benefit. Delivery system refers to the responsibility of central and local governments to provide LTC services for clients. Finance refers to financing methods in terms of the size of LTC budgets and the ratio of the public and private payment of LTC costs in the study.

OECD statistics and reports were mainly used for the data used in the study. When it was difficult to collect official data, information was collected by referring to the studies of individual researchers.

## 4. Research Results

Based on the LTC financing system, the analysis targeted 15 countries with tax-based systems, 6 countries with health insurance systems, and 5 countries with LTC insurance systems. Prior to the analysis, the general characteristics of 26 OECD countries were reviewed (Table 2). Table 2 classifies countries by their type of financing system and compares the elderly population ratio, the elderly dependency ratio, GDP per capita, health expenditure, and health status. As of 2020, the average OECD elderly population ratio was 17.5%, and the elderly dependency ratio was 0.27. In tax-based, health insurance-based, and LTC insurance-based countries, the average elderly population ratio was 19.1%, 18.9%, and 20.1%, respectively, and the elderly dependency ratio was 0.30, 0.29, and 0.31, which indicated no significant difference between groups. The OECD average GDP per capita was $47,510, while health expenditure was 10.4% of the GDP, and 46.5% of the elderly (aged 65 and over) answered good or very good regarding their health status. The average GDP of LTC insurance-based countries was $47,510, which was the highest among the three types, followed by tax-based countries ($45,459) and health insurance-based countries ($48,243). Health expenditure averages were 9.8% of GDP in LTC insurance-based countries and 10.5% in tax-based and health insurance-based countries. The health status was 49.5% for tax-based countries, 44.2% for health insurance-based countries, and 40.4% for LTC insurance-based countries.

In order to compare changes in elderly populations and elderly dependency ratios for each type, we reviewed the rates of increases and decreases in elderly populations and elderly dependency ratios in 2020 compared with those in 2010. The OECD average ratio of the elderly population increased by 21.1%, and the elderly dependency ratio increased by 24.5%. Tax, health insurance, and LTC insurance revealed no significant difference in the rate of increase in the proportion of the elderly population in each group.

The next section focuses on examining whether there are differences in LTC financing type through comparison-based averages.

### 4.1. Allocation: Selection Criteria (Selectivity and Universality), Level of Coverage

The selection criteria for LTC systems were largely classified into age, dependency status, and income. The scope of coverage varied from country to country. Table 3 compares the eligibility criteria for LTC systems in 26 OECD countries.

Most OECD countries do not explicitly impose age restrictions in the scope of the LTC system. However, Japan and Republic of Korea have age conditions for people under 65 years, such as those with age-related diseases, including dementia and stroke.

Dependency status standards, such as the activities of daily living and cognitive function, were applied as standards in most OECD countries that introduced LTC systems, except for a few Nordic countries, including Sweden and Denmark.

A means test is a representative criterion that distinguishes between selectivism and universalism. In most cases, tax-based countries provided public services only to certain classes, such as the low-income class. Except for Nordic countries, such as Sweden and Norway, most countries with tax and health insurance systems used the means-test as the selection criterion. Furthermore, countries with an LTC insurance model followed universalism because no standards were set for a means-test. Meanwhile, in Germany, a means-test was not the standard for basic service. However, the means test was used to select those who seek additional services other than basic services and to provide additional services [19].

Regarding the level of coverage, the rate of receiving official benefits through the LTC system among the 26 OECD countries constituted 5.8% of the population aged 15 years and above, but the difference in terms of the type of financing was insignificant. For elderly persons aged 65 years or older, who are the main targets of the LTC system, the overall coverage rate in OECD countries in 2019 was 12.7%, which was an increase from 12.4% in 2010. Home benefit (8.9%) had a higher coverage rate compared with institution benefit (3.8%). Compared with the rates in 2010, home benefit increased, and institution benefit decreased, which led to the conjecture that a change in policy toward community care was taking place (Table 4).

In terms of the type of financing, health insurance (14.0%) had the highest coverage rate in 2019, followed by LTC insurance (13.2%) and taxation (12.1%). In countries adopting a taxation system, the coverage rate decreased (12.7% → 12.1%) in 2019 compared with that in 2010. Furthermore, a decrease was observed in all types of benefits, including home and institution benefits. In countries adopting health insurance systems, the increase in coverage rates was the largest in 2019 compared with that in 2010 (11.5% → 14.0%). The coverage rates increased across all types of benefits, including home and institution benefits. In countries adopting LTC insurance systems, the overall coverage rate increased (12.6% → 13.2%) in 2019 compared with that in 2010. It was confirmed that the institution benefit decreased and that the home benefit increased (Table 4).

### 4.2. Benefits: Benefit Type and Amount of Benefits per Capita

LTC system benefits can be classified into cash benefit and in-kind benefit, and in-kind benefit is further classified into services and goods. For goods that mainly relate to the rental of welfare equipment, the proportions of expenditures and benefits are relatively inadequate.

European countries where social allowances have long been developed generally provide LTC benefits as cash benefits. Cash benefits are divided into benefits paid to informal caregivers, such as family members and relatives, and benefits paid to care recipients. Depending on the country, payments are made only to informal caregivers, only to the beneficiaries of care, or to everyone [21].

In-kind benefits refer to services and are mainly divided into home and institution benefits. In many countries that have introduced LTC systems, home benefits are prioritized over institutional benefits as they focus on community care, and efforts are being made to reduce the number of residents in institutions by allowing the recipients to stay their own homes as long as possible [22].

To examine differences based on the financing system, as indicated in Table 5, LTC services are classified into health services and social services. The proportions of expenditures are classified and examined by each type. Reviewing the average expenditure ratio of OECD countries, the expenditure ratio for institution benefits (65.7%) was about twice that of home benefits (34.4%). The average expenditure ratio of home benefits in countries adopting a taxation system was 40.7%, which was higher than that of home benefits in countries adopting health insurance systems (29.0%) and countries adopting LTC insurance systems (24.7%). The health insurance system (74.6%) had the highest ratio of institution benefits.

Figure 1 presents the absolute change in public expenditures of LTC allocated to institution and home care during 2011–2016 (or the closest year). Irrespective of the financing system, institution benefits decreased and home benefits increased in most countries.

To examine the number of benefits, we examined LTC expenditure per capita in each country and compared the amount of benefits in 2019 and 2010. Table 6 presents the value in which the amount of LTC benefit per capita in each country is converted into USD and the value in which per capita GDP is converted to the purchasing power of USD in each country. It also suggests the ratio of LTC benefit amounts and the amount of differences in relation to GDP purchasing power per capita.

LTC spending per capita in all 26 OECD countries in 2019 was $768.3, constituting 1.44% of GDP per capita. LTC insurance ($1034.2) had the highest per capita expenditure, followed by tax ($741.3) and health insurance ($614.3).

In the 2010 and 2019, per capita expenditure on LTC in OECD countries increased by an average of $270. The increase rate as a percentage of GDP per capita was 0.09%, and the expenditure rate also increased in relation to income, which confirmed the actual increase in expenditure. During the same period, countries with LTC insurance systems increased by $357 per capita, which was the highest cost increase at 0.18% of GDP per capita. Furthermore, countries with a taxation base had the lowest rate of increase at $271 per capita, which increased by 0.08% of GDP per capita. Countries with health insurance systems had $195 per capita, which increased by 0.02% of GDP per capita. Meanwhile, Republic of Korea had the lowest per capita LTC spending among countries with LTC insurance systems in 2019. Its per capita GDP ratio was 1.07%, which was the second lowest after Luxembourg. However, the increase in per capita GDP in 2019 compared to that in 2010 was 0.58%, which was the second highest after Sweden (2.02%).

### 4.3. Service Delivery: Central and Local Governments

The LTC service delivery system was classified into recipient selection, financial resources, and service provision, and the characteristics were analyzed by examining who takes responsibility between the central and local governments (Table 7). Although differences exist in administrative systems and names for each country, the names of the federal government, central government, and central government agencies were unified as the central government, and regions, local governments, and local government agencies were unified as local governments.

Regarding the responsibility for recipient selection, local governments set criteria for recipient selection in 11 of 19 OECD countries. However, in the remaining seven countries, the central governments set the criteria for the selection of recipients. In Poland, the central and local governments together set the criteria for the selection of recipients.

In most countries, both central and local governments were responsible for financial resources. Local governments in Denmark, Hungary, and the Netherlands assumed financial responsibility, whereas central governments in Republic of Korea, Austria, Ireland, and Australia assumed financial responsibility.

The main actors responsible for providing services were local governments. Of the 18 countries identified, 12 were responsible for providing services to local governments. However, in the Czech Republic, Slovakia, Australia, Germany, and Luxembourg, the central governments were responsible for providing services. In the case of the U.S., it was different across states.

### 4.4. Finance

#### 4.4.1. Budget Size

As of 2018, the average LTC expenditure as a percentage of GDP in 26 OECD countries was 1.47%. By type, LTC insurance-based countries had the highest LTC expenditure as a percentage of GDP at 1.75%, followed by countries with tax systems (1.54%) and countries with health insurance systems (1.13%) (Figure 2).

#### 4.4.2. Percentage of Public and Private Expenditure

Regarding household expenditure among total expenditure on LTC as a percentage of GDP, household expenditure in 26 OECD countries constituted an average of 0.22% of GDP. For each type, countries with a health insurance system had the highest household expenditure at 0.29%, followed by those with an LTC insurance system at 0.25% and those with a taxation system at 0.19% (Figure 3).

## 5. Conclusions

Based on Gilbert and Terrell’s policy analysis framework, 26 OECD countries that have introduced LTC systems were classified according to their financing system, and their characteristics were examined. The results for each analysis dimension are summarized as follows (Table 8).

In terms of allocation, the criteria for the selection of subjects, selectivity, and universality were examined. The results of examining the eligibility conditions for recipients indicated that the dependency criterion became the main criterion in most countries that introduced LTC systems, except for some Nordic countries. Exceptionally, the age criterion was applied only in Japan and Republic of Korea [26]. The means test was mainly applied by countries with taxation and health insurance systems. Taxation-based countries particularly provided services to the most vulnerable based on selectivism. However, countries with LTC insurance systems did not apply it uniformly.

In terms of benefits, the average major expenditure share of OECD countries of institution benefit (65.7%) was twice that of home benefit (34.4%). Following the analysis of the rankings for each benefit type, the ratio of home benefit expenditure was taxation (40.7%), health insurance (29.0%), and LTC insurance (24.7%). The ratio of institution benefit expenditure was health insurance (74.6%), LTC insurance (67.5%), and taxation (61.5%). The results of changes in LTC public expenditures between 2011 and 2016 indicated that, on average, home benefit increased, and institution benefit decreased in OECD countries. Regarding per capita benefit amount and per capita public expenditure cost, a country with the highest benefit amount was one with an LTC insurance system, followed by countries with taxation and health insurance systems.

In terms of delivery, we focused on the roles of central and local governments. In the end, no differences were observed in the type of financing. Responsibility for selecting recipients and providing services mainly rested with local governments, and financial responsibilities were mainly shared between central and local governments.

In terms of finance, the size of the budget and the ratio of public and private burdens were examined. In terms of budget size, as of 2018, the ratio of LTC expenditure to GDP was highest in a country with an LTC insurance system (1.75%), followed by those with taxation (1.54%) and health insurance (1.13%). The share of household spending in total LTC expenditure was 0.22% of the OECD average GDP, and health insurance system countries had the highest household expenditure (0.29%). This was followed by LTC insurance (0.25%) and taxation-based countries (0.19%).

Through this study, several characteristics were identified according to the three different financing systems. Firstly, the taxation system adopted by some countries provided services to the most vulnerable based on selectivism, and the coverage rate for elderly persons aged 65 years and above declined. In terms of benefits, the home benefit ratio was the highest, and the institution benefit ratio was the lowest. Furthermore, the private pay ratio was the lowest. In terms of finance, the budget had the highest rate of home benefit and the lowest rate of institution benefit.

Secondly, although the health insurance system was a form of insurance at the level of allocation, services were provided based on selectivism. The coverage rate was the highest, and the coverage rate steadily increased in both homes and institutions. In terms of benefits, the share of institution benefit expenditure was the highest, and the per capita amount of benefit was the lowest. In terms of finance, the budget size and growth rate as a percentage of GDP was the lowest, and the private pay ratio was the highest.

Finally, countries operating LTC insurance systems followed universalism in terms of allocation and had the highest coverage rate for the elderly aged 65 and above. It was the only type of system in which an increase in the number of home benefit recipients and an decrease in the number of institution benefit recipients were observed. In terms of benefit, the amount of per capita benefit was the highest, and in terms of finance, the budget-to-GDP ratio and increase rate were the highest. The share of public expenditure was the highest, and the rate of home benefit was the highest in the ratio of public and private benefits.

This study is different from previous studies in that it analyzed the characteristics of countries according to financing systems such as taxation, health insurance, and LTC insurance. In particular, the accomplishments of the study are that it identified differences that were difficult to find in previous studies by comparatively analyzing allocation, benefits, delivery, and finance through the research analysis framework of Gilbert and Terrell.

## 6. Discussion

In this study, it was discovered that there are many different LTC financial resources, different types of supplying entities, and different benefit application rates. The reason for these differences is that they chose to increase the number of private providers by introducing a quasi-market mechanism to long-term care services in order to increase cost efficiency. This means that the cost burden of LTC recipients could not be reduced, and sufficient financial support was not provided at the government level. Combining these factors, it becomes clear that the environment in which adequate LTC can be delivered affects LTC suppliers and the way financial resources are set up. Based on this, it is suggested that the expansion of the public sector, appropriate support, and reasonable regulation for LTC suppliers are necessary.

This study has a few limitations. Firstly, owing to the use of Gilbert and Terrell’s framework, only specific dimensions of long-term care were analyzed. Although it was inevitable to compare some aspects of long-term care, other key issues, such as workforce, were not explored due to the unique situations in each country. Second, the analysis of service delivery dimension was not performed sufficiently due to the limitations of accessing and securing relevant data. Thirdly, the comparison years are somewhat different due to different data sets we acquired. Finally, it was difficult to compare given the unique history of OECD countries. To overcome the limitations, it is necessary to conduct a comprehensive and large study by fully considering the historical and institutional conditions of each country. It is hoped that research will be further developed through follow-up studies.

## Figures and Tables

**Figure 1 healthcare-11-00206-f001:**
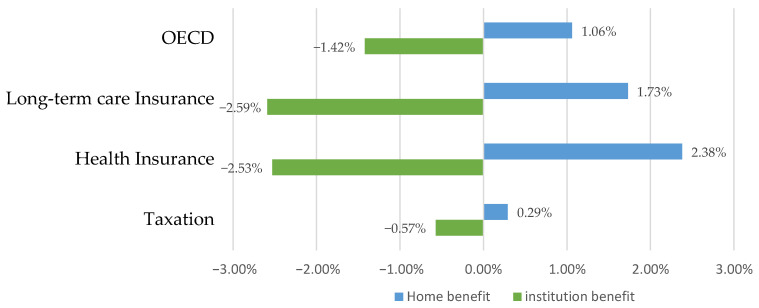
Changes in public expenditure for long-term care from 2011–2016 [23].

**Figure 2 healthcare-11-00206-f002:**
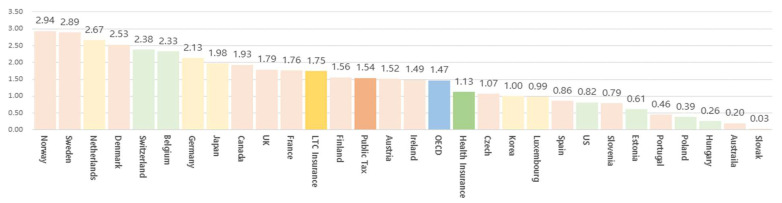
Status of long-term care expenditure as a percentage of GDP in 26 OECD countries (2019). Source: OECD Health Statistics [18].

**Figure 3 healthcare-11-00206-f003:**
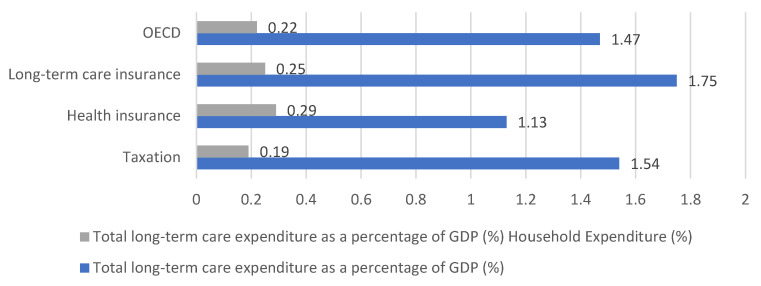
Total long-term care expenditure and household expenditure as a percentage of GDP [18,25].

**Table 1 healthcare-11-00206-t001:** Analysis framework of the study.

Dimension	Scope	Analysis Content
Allocation (Allocation)	Target system (Eligibility criteria)	- Target selection criteria (selectivity/universality) - Level of coverage (coverage rate of 65 years old and above)
Benefit (Provision)	Benefit system (Type of benefit)	- Form of benefit: ratio of home and institutional benefit - Benefit amount (per capita)
Delivery (Delivery)	Delivery system (Delivery method)	- Central and local authority responsibility
Finance (Finance)	Financial system (Financial system)	- Budget size: type, size, etc. - Ratio of public and private payment

**Table 2 healthcare-11-00206-t002:** Analysis of long-term care financing types by OECD country (as of 2020) [18].

No.	Country	Type of Financing (Main Resource)	Total Population (Million People)		Elderly Population Ratio (%)		Elderly Dependency Ratio		GDP (per Capita, USD PPPs)	Health Expenditure (%, Share of GDP)	Health Status (Good/Very Good, 65+)
			2020	Variation ^(1)^	2020	Variation	2020	Variation	2020	2020	2020
1	Sweden	Taxation	10.4	10.4%	20.1	9.7%	0.32	14.9%	49,491	11.5	59.4
2	Norway		5.4	10.0%	17.7	18.4%	0.27	20.4%	60,911	9.7	60.6
3	Finland		5.5	3.1%	22.5	30.3%	0.36	39.4%	44,724	9.6	51.4
4	Denmark		5.8	5.1%	20	20.9%	0.31	24.2%	51,493	10.5	57.1
5	Austria		8.9	6.6%	19.2	8.4%	0.29	10.1%	49,031	11.5	47
6	Czech Republic		10.7	1.7%	20	30.4%	0.31	43.4%	36,208	9.2	27.5
7	Slovakia		5.5	0.5%	16.8	36.2%	0.25	46.4%	32,283	9.2	23.6
8	Slovenia		2.1	2.5%	20.5	23.7%	0.32	33.0%	34,708	9.5	34.1
9	Portugal		10.3	−2.6%	22.3	20.6%	0.35	24.6%	30,512	10.5	13
10	Spain		47.4	1.7%	19.6	15.9%	0.3	19.7%	33,613	10.7	42.9
11	The UK		67.1	6.9%	18.6	14.0%	0.29	18.6%	39,788	12	57.1
12	France		67.3	4.0%	20.6	23.3%	0.33	29.6%	39,548	12.2	44.5
13	Ireland		5	9.3%	14.5	27.9%	0.22	32.4%	88,111	7.1	69
14	Australia		25.7	16.6%	16.3	20.4%	0.25	24.7%	48,094	10.6	73.8
15	Canada		38	11.8%	18	27.3%	0.27	33.5%	43,376	12.9	82.2
Tax-based country			315	5.8%	19.1	21.2%	0.3	26.8%	45,459	10.5	49.5
16	Estonia	Health Insurance	1.3	−0.1%	20.2	15.8%	0.32	23.0%	33,746	7.8	20.6
17	USA		329.5	6.5%	16.9	29.1%	0.26	33.7%	58,408	18.8	77.4
18	Belgium		11.5	5.6%	19.3	12.2%	0.3	15.6%	45,733	11.1	53.8
19	Hungary		9.8	−2.5%	20.1	20.5%	0.31	26.6%	30,404	7.3	21
20	Poland		38.4	−0.4%	18.4	36.9%	0.28	47.3%	31,179	6.5	22.4
21	Switzerland		8.6	10.4%	18.7	10.7%	0.28	13.6%	65,754	11.8	69.7
Health Insurance-based countries			399.1	5.6%	18.9	20.9%	0.29	25.4%	44,204	10.5	44.2
22	Germany	Long-term Care Insurance	83.2	1.7%	21.9	6.0%	0.34	8.5%	48,243	12.8	39.1
23	Netherlands		17.4	5.0%	19.6	27.1%	0.3	31.6%	51,522	11.1	62.2
24	Luxembourg		0.6	24.4%	14.6	4.6%	0.21	3.0%	106,383	5.8	55.5
25	Japan		125.7	−1.8%	28.8	25.0%	0.49	34.7%	40,604	11.1	25.1
26	Republic of Korea		51.8	4.5%	15.7	44.9%	0.22	46.8%	41,385	8.4	20
Long-term Care Insurance-based Countries			278.7	0.8%	20.1	21.5%	0.31	24.9%	57,627	9.8	40.4
OECD			1369	5.8%	17.5	21.1%	0.27	24.5%	47,510	10.4	46.5

^(1)^ Variation when compared with 2010. Explanation: missing values of expenditures in 2020 are substituted with those from the closest year.

**Table 3 healthcare-11-00206-t003:** Comparison of long-term care system eligibility criteria according to each OECD country [20].

No.	Classification	Country	Age Criteria	Dependency Criteria	Means Test Criteria
1	Taxation	Sweden	no	yes	no
2		Norway	no	-	no
3		Finland	no	yes	no
4		Denmark	no	no	no
5		Austria	no	yes	yes
6		Czech Republic	no	yes	no
7		Slovakia	no	yes	yes
8		Slovenia	no	yes	yes
9		Portugal	no	no	yes
10		Spain	no	yes	yes
11		The UK	no	yes	yes
12		France	no	yes	yes
13		Ireland	no	yes	yes
14		Australia	-	-	-
15		Canada	no	-	-
		Ratio	0.0%	81.8%	66.7%
16	Health Insurance (+Taxation)	Estonia	no	no	no
17		USA	no	-	yes
18		Belgium	no	yes	yes
19		Hungary	no	yes	yes
20		Poland	no	yes	yes
21		Switzerland	no	-	-
		Ratio	0.0%	75.0%	80.0%
22	Long-term Care Insurance (+Taxation)	Germany	no	yes	no
23		Netherlands	no	no	no
24		Luxembourg	no	yes	no
25		Japan	yes	yes	no
26		Republic of Korea	yes	yes	no
		Ratio	40.0%	80.0%	0.0%

**Table 4 healthcare-11-00206-t004:** Long-term care systems’ coverage rate according to OECD country (65 years old and above) (2010 and 2019).

Classification	Coverage Rate (65 Years Old and Above, %)					
	Institution		Home		Total	
	2019	2010	2019	2010	2019	2010
Taxation	3.6	4.2	8.4	8.6	12.1	12.7
Health Insurance	4.3	3.7	10.0	7.8	14.0	11.5
Long-term Care Insurance	3.8	4.1	9.0	8.5	13.2	12.6
OECD	3.8	4.1	8.9	8.4	12.7	12.4

**Table 5 healthcare-11-00206-t005:** Expenditure ratios by health and social long-term care benefit type [2].

Classification	Health Service					Social Service
		Institution	Weekly	Outpatient	Home	
Taxation	82.1%	61.5%	3.5%	11.5%	40.7%	17.9%
Health Insurance	90.3%	74.6%	1.0%	1.2%	29.0%	11.7%
Long-term Care Insurance	87.9%	67.5%	7.2%	1.5%	24.7%	12.1%
OECD	85.1%	65.7%	4.0%	6.4%	34.4%	15.5%

**Table 6 healthcare-11-00206-t006:** Long-term care expenditure per capita (USD, PPP) as a percentage of GDP per capita (2019) [18].

Classification	A. LTC Expenditure per Capita (USD)			B. GDP per Capita (USD PPP)			A/B × 100 (%)		
	2019	2010	Difference	2019	2010	Difference	2019	2010	Difference
Taxation	741.3	501	271	51,214	36,441	14,773	1.45%	1.37%	0.08%
Health Insurance	614.3	419	195	49,734	34,421	15,313	1.24%	1.22%	0.02%
LTC Insurance	1034.2	677	357	64,149	47,309	16,840	1.61%	1.43%	0.18%
OECD	768.3	516	270	53,360	38,065	15,295	1.44%	1.35%	0.09%

Explanation: missing values of expenditures in 2019 and 2010 are substituted with those from the closest year.

**Table 7 healthcare-11-00206-t007:** The role of central and local governments in service delivery systems [24].

No.	Financing	Country	Responsibility for Recipient Selection	Financial Responsibility	Responsibility for Service Provision
1	Taxation	Sweden	Local	Central and local	Local
2		Norway	Local	Central and local	Local
3		Finland	-	-	-
4		Denmark	Local	Local	Local
5		Austria	Central	Central	Central
6		Czech Republic	Central	Central and local	Central
7		Slovakia	-	-	-
8		Slovenia	-	-	-
9		Portugal	-	-	-
10		Spain	Local	Central and local	Local
11		The UK	Local	Central and local	Local
12		France	Central	Central and local	Local
13		Ireland	-	Central	-
14		Australia	Central	Central	Central
15		Canada	Local	Central and local	-
16	Health Insurance	Estonia	-	-	-
17		USA	Local	Central and local	Central and local
18		Belgium	-	-	-
19		Hungary	Local	Local	Local
20		Poland	Central and local	Central and local	Local
21		Switzerland	Local	Local	Local
22	Long-term Care Insurance	Germany	Central	Central and local	Central
23		Netherlands	Local	Local	Local
24		Luxembourg	Central	Central (sickness fund)	Central (sickness fund)
25		Japan	Local	Central and local	Local
26		Republic of Korea	Central	Central and local	Local

**Table 8 healthcare-11-00206-t008:** Research results.

Dimension	Scope	Analysis Content	Analysis Results
Allocation	Target system (qualifications)	Recipient selection criteria	Dependency criteria: common (except for some Nordic countries) Age criteria: Republic of Korea and Japan
		Selectivity/universality classification (the presence of a means test)	Universality: LTCI; selectivity: taxation and health insurance
		Level of coverage (65 years old and above)	Coverage rate: health insurance > LTCI > taxation Home and institution classification coverage rate - Taxation: total reduction - Health insurance: overall increase - LTCI: decrease in institutions and increase in homes
Benefit (Provision)	Benefit system (type of benefit)	Benefit type	Home and institution benefits (proportion of expenditure) - Home benefits: taxation > health insurance > LTCI - Institution benefits: health insurance > LTCI > taxation - Expenditure increase rate: increase in home benefit expenses
		Benefit amount (per capita)	- LTCI > taxation > health insurance
Delivery	Delivery system (delivery method)	Central/local	No difference by type - Responsibilities for recipient selection: local - Financial responsibilities: central and local - Responsibilities for providing services: local
Finance	Financial system	Budget size	LTCI > taxation > health insurance
		Public/private pay ratio	Public: taxation > LTCI > health insurance Individual: health insurance > LTCI > taxation

## Data Availability

Not applicable.

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
