# Peer review of "Comparative Analysis of Long-Term Care in OECD Countries: Focusing on Long-Term Care Financing Type"

_healthcare, 2023, doi:10.3390/healthcare11020206_

Round 1
Reviewer 1 Report
Drawing primarily on OECD data, this paper uses Gilbert and Terrell’s categories of allocation, benefit provision, delivery, and finance to compare countries with long-term care systems based on taxation, health insurance and long-term care insurance. Although the paper provides some useful comparisons, it requires significant revision to clarify the comparisons and draw out conclusions.
The introduction (30-61) sets out several generalizations that could be challenged. For example, major changes in long-term care took place in the early 1970s in a number of countries while this introduction suggests it all starts in the 1990s. (If it did start in the 1990s, why do the data used in this paper start in 2010?). There is no need for these generalizations. The paper could just say that long-term care has been receiving increasing attention.
Section 2 lists three kinds of financing systems (which I would not describe as theory) but it does not explain clearly the distinction among them or how these terms are understood. It is not clear how tax-based systems differ from health insurance and long-term care insurance. Health insurance systems, as commonly understood, may be primarily funded through taxes. Given these distinctions are at the core of the comparisons, it is important to clarify what is meant.
Section 3 provides some very brief description of Gilbert and Terrell’s categories, but more is needed on how each of these is defined and measured. While it is necessary in making such broad comparisons to overcome some complexity, there needs to be more discussion of the complexity that may be hidden. It would be useful, for example, to explain more fully than Table 1 does how selectivity and universality are understood. Norway, and Sweden do not base entry on ability to pay or specific age criteria but do charge fees. This is also the case with Canada, although the Table seems to suggest entry is means tested in Canada. Selecting out Nordic countries, as for example is the case in Table 10, implies that there is uniformity elsewhere which is not necessarily the case.
It would also be useful to know what 2010 was selected as the comparator year in Table 2. In Table 3, it is not entirely clear what the x and o mean. In section 4, the distinction between health and social services is not clear.
It would be useful to know why there is no country by country comparison for tables on benefit type, public expenditure, per capita expenditures and households when there is in earlier tables.
In the discussion, it would be useful to talk about patterns rather than make statements like “the taxation system adopted by some countries provided services to the most vulnerable based on selectivism (351-352).
It would also be useful to have some discussion about the implications of these patterns. Why are they useful to investigate?
Author Response
Reviewer 1
Drawing primarily on OECD data, this paper uses Gilbert and Terrell’s categories of allocation, benefits provision, delivery, and finance to compare countries with long-term care systems based on taxation, health insurance, and long-term care insurance. Although the paper provides some useful comparisons, it requires significant revision to clarify the comparisons and draw out conclusions.
The introduction (30-61) sets out several generalizations that could be challenged. For example, major changes in long-term care took place in the early 1970s in a number of countries while this introduction suggests it all starts in the 1990s. (If it did start in the 1990s, why do the data used in this paper start in 2010?). There is no need for these generalizations. The paper could just say that long-term care has been receiving increasing attention.
Thank you for your review comments. Added as below as you advised. The reason this study started with data from the 2010s is that it represents the results after institutional reform in the 1990s.
The increasing demand for the LTC for the elderly and the accompanying financial pressures have led to institutional reforms in most countries that have introduced LTC systems [3] The reason this study started with data from the 2010s is that it represents the results after institutional reform in the 1990s.
Section 2 lists three kinds of financing systems (which I would not describe as theory) but it does not explain clearly the distinction among them or how these terms are understood. It is not clear how tax-based systems differ from health insurance and long-term care insurance. Health insurance systems, as commonly understood, may be primarily funded through taxes. Given these distinctions are at the core of the comparisons, it is important to clarify what is meant.
Thank you for your review comments. Added as below. The explanation may not be sufficient, but We have tried to differentiate between the three types of finance.
The long-term care system is connected to the existing social security system in each country to form a financial system. From a publicity perspective, several OECD countries perceive long-term care services as universal services that are paid for by taxation (OECD, 2017). (U. Schneider, 1999)
However, countries with social insurance systems like Germany, Korea, Japan, Luxembourg, and the Netherlands also offer long-term care insurance. It is a system in which users contribute like a pension. These countries pursued the continuity of the welfare state while reducing the financial burden due to taxation, expanding the recipients of long-term care services, expanding the diversity of benefits, and expanding supply in the private sector. On the other hand, the United States, Belgium, and Hungary are operating long-term care systems within health insurance, and they are also promoting the role of the private sector.
Section 3 provides some very brief descriptions of Gilbert and Terrell’s categories, but more is needed on how each of these is defined and measured. While it is necessary in making such broad comparisons to overcome some complexity, there needs to be more discussion of the complexity that may be hidden. It would be useful, for example, to explain more fully than Table 1 does how selectivity and universality are understood. Norway and Sweden do not base entry on the ability to pay or specific age criteria but do charge fees. This is also the case with Canada, although the Table seems to suggest entry is means tested in Canada. Selecting out Nordic countries, as for example is the case in Table 10, implies that there is uniformity elsewhere which is not necessarily the case.
Thank you for your review comments. Added as below.
to compare the system and outcome of the policy of different countries, since it suggested a useful framework to classify and analyze essential elements of policy design into four areas of allocation, benefits, delivery system, and finance. These are defined given the aim of the study as follows: the allocation means to whom LTC service will be provided. Selectivism is based on the eligibility criteria of means(or/with income) and the degree of need for care(such as Activities of Daily Living). It tends to provide public-based long-term services mainly to the poor. However, universalism is only based on the degree of need for care, regardless of the financial situations of the person.
It would also be useful to know what 2010 was selected as the comparator year in Table 2. In Table 3, it is not entirely clear what the x and o mean. In section 4, the distinction between health and social services is not clear.
Thank you for your review comments. Added as below. The expression of the x and o has also been changed to yes and no.
Long-term care (LCT) expenditure is divided into LCT health and LCT social expenditure based on the System of Health Accounts (SHA) classification system. While the latter refers to health-realized social care as HC.R.6, the former refers to long-term nursing and personal care as an HC.3 category of SHA. As shown in the table, some countries have only health spending, while others include social spending as well.
It would be useful to know why there is no country-by-country comparison for tables on benefit type, public expenditure, per capita expenditures, and households when there are earlier tables.
Thank you for your review comments. Added as below.
In the next section, we will focus on examining whether there are differences in LTC financing type through comparison based on the average.
In the discussion, it would be useful to talk about patterns rather than make statements like “the taxation system adopted by some countries provided services to the most vulnerable based on selectivism (351-352).
Thank you for your review comments. Added as below.
It was mentioned in the text, but it wasn't enough.
Especially, the taxation-based countries provided services to the most vulnerable based on selectivism
It would also be useful to have some discussion about the implications of these patterns. Why are they useful to investigate?
Thank you for your review comments. Added as below.
In this study, it was discovered that there are many different LTC financial resources, different types of supplying entities, and different benefit application rates. The reason for this difference in type is that they choose to increase the number of private providers by introducing a quasi-market mechanism to long-term care services in order to increase cost efficiency. This means that the cost burden of LTC recipients could not be reduced, and sufficient financial support was not provided at the government level. Combining these factors, it becomes clear that the environment in which adequate LTC can be delivered affects the LTC supplier and the way financial resources are set up. Based on this, it was suggested that the expansion of the public sector, appropriate support, and reasonable regulation for LTC suppliers are necessary.
In addition, the conclusion was completely revised and the following explanations were added.
This study has a few limitations as follows: firstly, owing to the use of Gilbert and Terrell’s framework, specific dimensions of long-term care were analyzed only. Although it was inevitable to compare some aspects of long-term care, other key issues such as workforce were not explored given the unique situations of each country. Second, the analysis of service delivery dimension was not performed sufficiently due to the limitations of access and securing of data. Thirdly, the comparison years are somewhat different owing to the different data sets which we acquire. Finally, it was difficult to compare given the unique history of the OECD countries. To overcome the limitations, it is necessary to conduct comprehensive and big-size research by fully considering the historical and institutional conditions of each country. It is hoped that research will be further developed through follow-up studies.

Reviewer 2 Report
Dear editor
Thanks a lot for assigning this article to me for peer-review. As the authors tried to mention at the Introduction section- Theoretical Background, the article is novel from the comprehensiveness of the comparison and the methodological approach. There are some points mentioned below that would help improve the article`s understandability particularly for international readers:
1- A brief description about the OECD countries from their health index, GDP per capita and health expenditures can be helpful for those readers from middle east, northern Africa and Latin America who may not be a member of the OECD.
2- A brief clarifications about the characteristics and the pros and cons of each of the three studied categories; tax-based, social insurance-based, and private insurance-based models, can be helpful in theoretical background to show the significance of the topic.
3- I am wondering where the authors clarify the aim of their study regardless of their emphasis on the novelty and differences among the present study compared to the available knowledge, the aim should be mentioned and developed clearly.
4- The method section needs to be improved. There isn`t any clarification about data collection and analysis instruments. It is vague whether the authors have used descriptive analysis or summative analysis of the data may be used (based on the results presented in table 10). At the same time, there is no descriptions about the validity, credibility and reliability of the method used and data analysis.
5- There is some sort of mixing up between the discussion and the conclusion sections that could lead to misunderstanding. The final discussion about the several characteristics of the different three financing systems, are mostly appropriate as the main results or conclusion of this comparative study. In the discussion section the readers expect some interpretations of the results and comparison with available knowledge as well as justification of the differences. The implications of the study for health policymakers of the OECD countries and those in other regions are important.
6- The limitations of the study are not considered. Here referring to the robustness of the comparative study from the methodological perspective can be useful.
7- Due to the presenting many tables, it is suggested to illustrate the Analysis framework of the study as a figure. Some other tables could be merged for more comprehensiveness. Meanwhile presenting table 10 ain a conclusion section can be misleading.
Author Response
Reviewer 2
Dear editor
Thanks a lot for assigning this article to me for peer-review. As the authors tried to mention at the Introduction section- Theoretical Background, the article is novel from the comprehensiveness of the comparison and the methodological approach. There are some points mentioned below that would help improve the article`s understandability particularly for international readers:
1- A brief description about the OECD countries from their health index, GDP per capita and health expenditures can be helpful for those readers from middle east, northern Africa and Latin America who may not be a member of the OECD.
Thank you for your review comments. Added as below as you advised.
Table 2. Analysis of long-term care financing types by OECD country (as of 2020)
|
No. |
Country |
Type of Financing (main resource) |
Total population (million people) |
Elderly population ratio (%) |
Elderly dependency ratio |
GDP (per capita, PPPs, USD) |
Health Expenditure (%, share of GDP) |
Health Status (good/very good, 65+) |
|||
|
2020 |
Variation1) |
2020 |
Variation |
2020 |
Variation |
2020 |
2020 |
2020 |
|||
|
1 |
Sweden |
Taxation |
10.4 |
10.4% |
20.1 |
9.7% |
0.32 |
14.9% |
49,491 |
11.5 |
59.4 |
|
2 |
Norway |
5.4 |
10.0% |
17.7 |
18.4% |
0.27 |
20.4% |
60,911 |
9.7 |
60.6 |
|
|
3 |
Finland |
5.5 |
3.1% |
22.5 |
30.3% |
0.36 |
39.4% |
44,724 |
9.6 |
51.4 |
|
|
4 |
Denmark |
5.8 |
5.1% |
20 |
20.9% |
0.31 |
24.2% |
51,493 |
10.5 |
57.1 |
|
|
5 |
Austria |
8.9 |
6.6% |
19.2 |
8.4% |
0.29 |
10.1% |
49,031 |
11.5 |
47 |
|
|
6 |
Czech Republic |
10.7 |
1.7% |
20 |
30.4% |
0.31 |
43.4% |
36,208 |
9.2 |
27.5 |
|
|
7 |
Slovakia |
5.5 |
0.5% |
16.8 |
36.2% |
0.25 |
46.4% |
32,283 |
9.2 |
23.6 |
|
|
8 |
Slovenia |
2.1 |
2.5% |
20.5 |
23.7% |
0.32 |
33.0% |
34,708 |
9.5 |
34.1 |
|
|
9 |
Portugal |
10.3 |
-2.6% |
22.3 |
20.6% |
0.35 |
24.6% |
30,512 |
10.5 |
13 |
|
|
10 |
Spain |
47.4 |
1.7% |
19.6 |
15.9% |
0.3 |
19.7% |
33,613 |
10.7 |
42.9 |
|
|
11 |
The UK |
67.1 |
6.9% |
18.6 |
14.0% |
0.29 |
18.6% |
39,788 |
12 |
57.1 |
|
|
12 |
France |
67.3 |
4.0% |
20.6 |
23.3% |
0.33 |
29.6% |
39,548 |
12.2 |
44.5 |
|
|
13 |
Ireland |
5 |
9.3% |
14.5 |
27.9% |
0.22 |
32.4% |
88,111 |
7.1 |
69 |
|
|
14 |
Australia |
25.7 |
16.6% |
16.3 |
20.4% |
0.25 |
24.7% |
48,094 |
10.6 |
73.8 |
|
|
15 |
Canada |
38 |
11.8% |
18 |
27.3% |
0.27 |
33.5% |
43,376 |
12.9 |
82.2 |
|
|
Tax-based country |
315 |
5.8% |
19.1 |
21.2% |
0.3 |
26.8% |
45,459 |
10.5 |
49.5 |
||
|
16 |
Estonia |
Health Insurance |
1.3 |
-0.1% |
20.2 |
15.8% |
0.32 |
23.0% |
33,746 |
7.8 |
20.6 |
|
17 |
USA |
329.5 |
6.5% |
16.9 |
29.1% |
0.26 |
33.7% |
58,408 |
18.8 |
77.4 |
|
|
18 |
Belgium |
11.5 |
5.6% |
19.3 |
12.2% |
0.3 |
15.6% |
45,733 |
11.1 |
53.8 |
|
|
19 |
Hungary |
9.8 |
-2.5% |
20.1 |
20.5% |
0.31 |
26.6% |
30,404 |
7.3 |
21 |
|
|
20 |
Poland |
38.4 |
-0.4% |
18.4 |
36.9% |
0.28 |
47.3% |
31,179 |
6.5 |
22.4 |
|
|
21 |
Switzerland |
8.6 |
10.4% |
18.7 |
10.7% |
0.28 |
13.6% |
65,754 |
11.8 |
69.7 |
|
|
Health Insurance-based countries |
399.1 |
5.6% |
18.9 |
20.9% |
0.29 |
25.4% |
44,204 |
10.5 |
44.2 |
||
|
22 |
Germany |
Long-term care Insurance |
83.2 |
1.7% |
21.9 |
6.0% |
0.34 |
8.5% |
48,243 |
12.8 |
39.1 |
|
23 |
Netherlands |
17.4 |
5.0% |
19.6 |
27.1% |
0.3 |
31.6% |
51,522 |
11.1 |
62.2 |
|
|
24 |
Luxembourg |
0.6 |
24.4% |
14.6 |
4.6% |
0.21 |
3.0% |
106,383 |
5.8 |
55.5 |
|
|
25 |
Japan |
125.7 |
-1.8% |
28.8 |
25.0% |
0.49 |
34.7% |
40,604 |
11.1 |
25.1 |
|
|
26 |
South Korea |
51.8 |
4.5% |
15.7 |
44.9% |
0.22 |
46.8% |
41,385 |
8.4 |
20 |
|
|
Long-term care Insurance-based countries |
278.7 |
0.8% |
20.1 |
21.5% |
0.31 |
24.9% |
57,627 |
9.8 |
40.4 |
||
|
OECD |
1369 |
5.8% |
17.5 |
21.1% |
0.27 |
24.5% |
47,510 |
10.4 |
46.5 |
||
|
1) Variation comparing to 2010 Data: OECD Health Statistics [19]. Explanation: Missing values of expenditures in 2020 are substituted from those of the closest year |
|||||||||||
2- A brief clarifications about the characteristics and the pros and cons of each of the three studied categories; tax-based, social insurance-based, and private insurance-based models, can be helpful in theoretical background to show the significance of the topic.
Thank you for your review comments. Added as below.
The long-term care system is connected to the existing social security system in each country to form a financial system. From a publicity perspective, several OECD countries perceive long-term care services as universal services that are paid by taxation [2].
However, countries with social insurance systems like Germany, Korea, Japan, Luxembourg, and the Netherlands also offer long-term care insurance. It is a system in which users contribute like a pension. These countries pursued the continuity of the welfare state while reducing the financial burden due to taxation, expanding the recipients of long-term care services, expanding the diversity of benefits, and expanding supply in the private sector. On the other hand, the United States, Belgium, and Hungary are operating long-term care systems within health insurance, and they are also promoting the role of the private sector [8].
3- I am wondering where the authors clarify the aim of their study regardless of their emphasis on the novelty and differences among the present study compared to the available knowledge, the aim should be mentioned and developed clearly.
Thank you for your review comments. We did not write much in terms of quantity, but we emphasized it as follows.
The purpose of this study is to derive major characteristics by finding commonalities and differences of each type between LTC financing system
4- The method section needs to be improved. There isn`t any clarification about data collection and analysis instruments. It is vague whether the authors have used descriptive analysis or summative analysis of the data may be used (based on the results presented in table 10). At the same time, there is no descriptions about the validity, credibility and reliability of the method used and data analysis.
Thank you for your review comments. Added as below.
We mentioned it in the text, but it wasn't enough. Highlighted below.
to compare the system and outcome of policy of different countries, since it suggested an useful framework to classify and analyze essential elements of policy design into four areas of allocation, benefits, delivery system, and finance. These are defined given the aim of the study as follows: the allocation means to whom LTC service will be provided. Selectivism is based on the eligibility criteria of means(or/with income) and the degree of need of care(such as Activities of Daily Living). It tends to provide public-based long-term services mainly to the poor. However, universalism is only based on the degree of need of care, regardless of financial situations of the person.
For the data used in the study, OECD statistics and reports were mainly used. When it was difficult to collect official data, information was collected by referring to the studies by individual researchers
5- There is some sort of mixing up between the discussion and the conclusion sections that could lead to misunderstanding. The final discussion about the several characteristics of the different three financing systems, are mostly appropriate as the main results or conclusion of this comparative study. In the discussion section the readers expect some interpretations of the results and comparison with available knowledge as well as justification of the differences. The implications of the study for health policymakers of the OECD countries and those in other regions are important.
The part you said was moved to the conclusion part and added to the discussion part as follows.
In this study, it was discovered that there are many different LTC financial resources, different types of supplying entities, and different benefit application rates. The reason for this difference in type is that they chose to increase the number of private providers by introducing a quasi-market mechanism to long-term care services in order to increase cost efficiency. This means that the cost burden of LTC recipients could not be reduced, and sufficient financial support was not provided at the government level. Combining these factors, it becomes clear that the environment in which adequate LTC can be delivered affects the LTC supplier and the way financial resources are set up. Based on this, it was suggested that the expansion of the public sector, appropriate support, and reasonable regulation for LTC suppliers are necessary.
6- The limitations of the study are not considered. Here referring to the robustness of the comparative study from the methodological perspective can be useful.
Thank you for your review comments. Added as below.
This study has a few limitations as follows: firstly, owing to the use of Gilbert and Terrell’s framework, specific dimensions of long-term care were analyzed only. Although it was inevitable to compare some aspects of long-term care, other key issues such as workforce were not explored given the unique situations of each country. Second, the analysis of service delivery dimension was not performed sufficiently due to the limitations of access and securing of data. Thirdly, the comparison years are somewhat different owing to the different data set which we acquire. Finally, it was difficult to compare given the unique history of the OECD countries. To overcome the limitations, it is necessary to conduct a comprehensive and big size research by fully considering the historical and institutional conditions of each country. It is hoped that research will be further developed through follow-up studies.
7- Due to the presenting many tables, it is suggested to illustrate the Analysis framework of the study as a figure. Some other tables could be merged for more comprehensiveness. Meanwhile presenting table 10 ain a conclusion section can be misleading.
Thank you for your review comments. Added as below.
It was reflected in the text, but not enough.
Table 6. Changes in public expenditure for long-term care in 2011–2016
Data: OECD [23]
Figure 9. Total long-term care expenditure and household expenditure as a percentage of GDP
Source: OECD [26], OECD Health Statistics [19].
